# Position: Prompts for Public-Sector LLMs Should Be Governed as Commons

**Rashid Mushkani** [1,2]

## Abstract

This position paper argues that prompts used to deploy large language models (LLMs) in public-sector settings should be treated as governed artefacts rather than private, transient inputs. Prompts encode role instructions, decision framings, and value claims; prompt choice can materially shift outputs even when model weights and input records are held fixed. Existing governance tools, including model and dataset documentation, organisation-level policies, and post-training alignment, rarely make the *local* prompt collections used in deployment transparent, contestable, or auditable. We propose *Prompt Commons*: a versioned, community-maintained repository of prompt templates with provenance metadata, licensing, and moderation logs. Using a pilot dataset collected with community partners in a large North American city (443 human prompts; 3,317 after augmentation), we illustrate three governance states (open, curated, veto-enabled) and a negotiation-oriented ensemble method that aggregates stakeholder prompts into compromise recommendations. We close with falsifiable implications and an evaluation agenda for prompt-layer governance.

## 1. Introduction

Public institutions are experimenting with large language models (LLMs) for drafting text, summarising records, supporting service triage, and producing materials for planning and engagement. In these deployments, prompt templates are configuration: they specify roles, audiences, and what counts as a relevant consideration. When a template becomes the default for a recurring workflow, it acts as a de facto policy instrument, shaping tone, prioritisation, and trade-off framing even when model weights and input

records are held fixed.

Existing governance discussions in machine learning largely locate accountability in artefacts upstream or downstream of prompting. Documentation practices such as Model Cards and Datasheets aim to make model behaviour and dataset properties legible (Mitchell et al., 2019; Gebru et al., 2021). Post-training alignment aims to steer model behaviour via feedback, instruction tuning, or constitutional constraints (Christiano et al., 2017; Ouyang et al., 2022; Bai et al., 2022), and vendors publish platform policies. These mechanisms matter, but they typically do not make the prompt collections used in particular public deployments visible, contestable, or reproducible.

Urban AI research suggests that these gaps are salient in cities and local administrations, where deployments are shaped by local mandates, procurement constraints, and contested value trade-offs (Xia et al., 2025; Zhu & Liu, 2025; Yigitcanlar et al., 2023; Allam & Dhunny, 2019; Kirwan & Fu, 2020). In such settings, prompt templates often circulate informally across teams, contractors, or vendors, outside formal policy review. When a single prompt becomes the default, its embedded assumptions can be mistaken for properties of the model or the policy question, even when they originate from a narrow authorship base and remain editable.

**Position: Prompts used in public-sector LLM deployments should be treated as governed artefacts and maintained in community-managed prompt commons with versioning, licensing, and accountable moderation.** We refer to this arrangement as *Prompt Commons*. It implies testable predictions: (i) governed prompt releases shift observable output distributions under fixed models and inputs, (ii) governance procedures shift operational metrics such as time-to-remediation under specified workflows, and (iii) auditability improves, measured by traceability from outputs to prompt versions and reproducibility under logged configurations.

The paper advances this position in four steps. First, it clarifies why prompt configuration constitutes a distinct governance surface and why treating prompts as transient inputs limits auditability. Second, it specifies Prompt Commons as a repository-and-process design grounded in commons governance (Ostrom, 1990; Carlisle & Gruby, 2019) and in open-source practice (O'Mahony & Ferraro, 2007; The

[1]Université de Montréal, Montréal, Canada [2]Mila – Québec AI Institute, Montréal, Canada. Correspondence to: Rashid Mushkani <rashidmushkani@gmail.com>.

*Proceedings of the 43rd International Conference on Machine Learning*, Seoul, South Korea. PMLR 306, 2026. Copyright 2026 by the author(s).

Linux Foundation, 2020). Third, it reports an illustrative pilot testbed to show how governance states can be instantiated and measured, without claiming general effectiveness. Fourth, it addresses alternative views and outlines actions for researchers, model providers, and public institutions.

## 2. Prompts as a Governance Surface

### 2.1. Prompt sensitivity

LLMs can adapt to new tasks with limited task-specific data (Brown et al., 2020), and prompting techniques structure these adaptations by controlling instruction, context, and output formats (Liu et al., 2025). Methods such as chain-of-thought prompting can shift the form and apparent completeness of reasoning (Wei et al., 2022), while explanation prompting can alter the style and perceived justification of outputs (Rajani et al., 2019). These findings are often discussed in terms of accuracy and task performance. In public-sector use, the same sensitivity has a governance implication: prompting choices can shift which considerations are presented as relevant, how uncertainty is communicated, and which trade-offs are treated as admissible.

The framing role of prompts interacts with evidence that LLMs and alignment artefacts can reflect systematic political or ideological leanings (Liu et al., 2022; Bang et al., 2024; Fulay et al., 2024). A prompt can amplify or suppress such tendencies by setting constraints on what a "balanced" answer should include, by selecting which stakeholders are represented, or by specifying what constitutes harm. Prompt governance is therefore not reducible to "prompt engineering" as an optimisation activity. It is a procedural layer for negotiating, documenting, and revising deployment-time value framings that would otherwise remain implicit.

### 2.2. Why existing governance artefacts are insufficient

Documentation approaches are designed to support transparency across model releases and datasets (Mitchell et al., 2019; Gebru et al., 2021). They are less informative about how a model is configured for a particular deployment, where prompt templates can encode local policy framings and decision thresholds. Post-training alignment provides system-wide constraints (Christiano et al., 2017; Ouyang et al., 2022; Bai et al., 2022), but public-sector deployments require local variation across domains, jurisdictions, and communities, and platform policies rarely record local prompt edits. Organisational practice can therefore bypass both documentation and alignment intentions when staff adapt prompts without review logs.

Critical perspectives on scale and centralisation emphasise that model design choices, training corpora, and deployment incentives shape harms and accountability (Bender et al., 2021). Prompt Commons is not proposed as an alternative to these concerns. It treats the prompt layer as a place where local institutions can document how they apply a model within a specific mandate, and where affected groups can inspect and contest the normative assumptions encoded in prompt templates.

## 3. Prompt Commons: Design & Governance

### 3.1. Repository structure

Prompt Commons maintains prompts and prompt collections as plain text with version identifiers. Each prompt entry records provenance fields often missing in deployment practice: author group(s) (e.g., seniors, disability advocacy, LGBTQ+, ethnic or religious minority), locale (e.g., neighbourhood, corridor, or administrative unit), intended value claim (e.g., accessibility, biodiversity, safety), consent and access level, and an auditable change log. Changes are proposed and discussed through issues or pull requests, and merged with recorded rationales and timestamps. This structure is intended to make prompts reproducible and contestable without treating any single prompt as a default that generalises across all contexts.

### 3.2. What Prompt Commons enables beyond existing practice

Model Cards and Datasheets document properties of models and datasets (Mitchell et al., 2019; Gebru et al., 2021), but they do not capture the deployment-time prompt templates that operational teams actually use, nor do they provide change-controlled review for prompt edits. Post-training alignment constrains general model behaviour (Christiano et al., 2017; Ouyang et al., 2022; Bai et al., 2022), but it cannot resolve local, task-specific framing choices, and it rarely exposes prompt provenance.

Existing prompt repositories make sharing easier, but they typically lack governance primitives required for public deployment: (i) versioned prompt *releases* tied to a deployment configuration, (ii) enforceable contribution and review rules, (iii) veto, quarantine, and appeal with logged resolutions, (iv) consent, withdrawal, and licensing metadata, and (v) audit logs that link an output to the exact prompt version and governance state. Prompt Commons treats these primitives as first-class, repository-enforceable objects.

### 3.3. Governance states

To operationalise the position, we distinguish three governance states implementable in common repository tooling. In an *open* state, any authenticated contributor can propose prompts, with moderation limited to spam and basic safety checks. In a *curated* state, maintainers enforce metadata completeness, inclusion constraints across groups and top-

*Table 1.* Governance states in Prompt Commons, expressed as enforceable repository rules.

| State | Enforceable rule |
| --- | --- |
| Open | Any authenticated contributor may propose prompts; maintainers remove spam and unsafe content; all changes are traceable via commits and issues. |
| Curated | Merge requires a maintainer review; prompts must include required provenance fields; releases satisfy published inclusion constraints (minimum coverage across groups/locales) and a checklist. |
| Veto-enabled | Curated plus a formal "quarantine" mechanism: designated representative organisations can file a veto record that blocks release until a time-boxed review and resolution (accept, modify, or reject) is logged. |

ics, and a published merge checklist. In a *veto-enabled* state, representative organisations can flag prompts or outputs as harmful, quarantining them for time-boxed review with appeal. The aim is to separate who may propose prompts from which prompts are endorsed for a deployment.

This design adapts commons principles (Ostrom, 1990) to a digital repository setting, and it uses polycentric structure to allow neighbourhood-level prompt sets to federate into citywide collections (Carlisle & Gruby, 2019). It also draws on empirical work on moderation labour and governance legitimacy (Li et al., 2022; Cao et al., 2024; Tabassum et al., 2024; Weld et al., 2024).

Digital-commons adaptations of Ostrom's framework for governing shared resources emphasise how boundaries, monitoring, and conflict resolution can be operationalised through repository workflows and documented practices (Mozilla Foundation, 2021; Dulong de Rosnay & Stalder, 2020).

### 3.4. Negotiation-oriented prompt aggregation

Prompt Commons is intended to support negotiation rather than to collapse disagreement into a single prompt. One operational pattern is to treat prompts as proposals and to combine them using an explicit aggregation prompt that instructs the model to surface differences and propose a compromise. This approach aligns with work on democratic alignment and aggregation, which emphasises that procedural rules shape outcomes (Huang et al., 2025; Conitzer et al., 2024). In Prompt Commons, the "aggregation rule" is not only a mathematical operator but also a documented artefact that can be debated and revised alongside other prompts.

### 3.5. Licensing

Prompts are text, but licensing governs reuse across agencies and vendors, attribution, and whether derivatives remain open. A baseline is Creative Commons (CC BY 4.0 or BY–SA 4.0) (Creative Commons, 2013b;a; 2025). For derived artefacts that operationalise prompts (wrappers, tools), use-restricted licences such as OpenRAIL can prohibit specified harms while allowing reuse (Muñoz Ferrandis, 2022; RAIL Initiative, 2023; Keller & Bonato, 2023a;b; BigCode Project, 2023; ifrOSS, 2024). Work on responsible-AI licensing recommends standardised customisation to limit ambiguity while enabling context-sensitive guardrails (McDuff et al., 2024; Contractor et al., 2022). In Prompt Commons, licensing is negotiated: communities may accept attribution-only reuse while selecting stronger conditions for scaled deployments.

## 4. Pilot Illustration

### 4.1. Data and participants

We convened community partners via local civil-society organisations (disability advocacy, immigrant support, seniors' services, women's groups, LGBTQ+ organisations, neighbourhood associations), alongside urban practitioners and a national cultural institution. Figure 1 shows self-declared participant categories by age group. Participants consented to contribute de-identified prompt templates under a community licence, with withdrawal requests logged via a pseudonymous token; prompts were screened for personally identifiable information. Participants authored 443 prompts describing urban scenes and values; we augmented to 3,317 via de-duplication, value-preserving paraphrase (up to five per prompt with an instruction-tuned paraphraser), and scenario expansion. [1]

Human-authored prompts average 22.6 words (median 19) versus 31.7 after augmentation; vocabulary entropy rises from 7.53 to 8.39 bits. Common content tokens include *street*, *park*, and *trees*. Equity-related tokens appear with measurable frequency: *wheelchair* (7.6 %), *metro* (7.3 %), and *biodiversity* (4.7 %); *LGBTQ+* and *Indigenous* appear less often (0.7 % to 1.4 %). These descriptive frequencies indicate how representation varies with recruitment practice and motivate inclusion rules and audit metrics.

### 4.2. Evaluation protocol

The pilot evaluation is an illustrative, falsifiable testbed for prompt governance, not a claim of general effects. We fix one instruction-tuned chat LLM via API (temperature 0, top-$p$ 1, max 256 tokens) and hold inputs, rubric, and de-

---

[1]Organisation names are omitted to protect partner confidentiality.

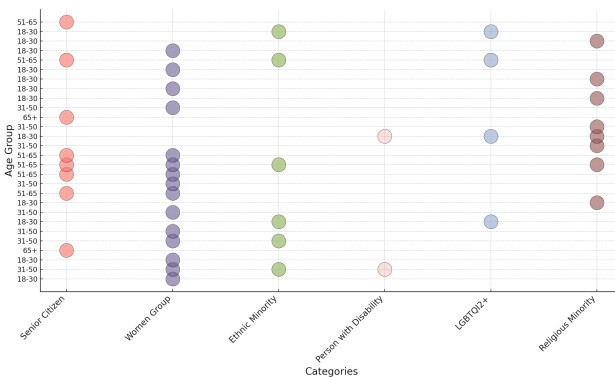

*Figure 1.* Self-declared participant categories by age group.

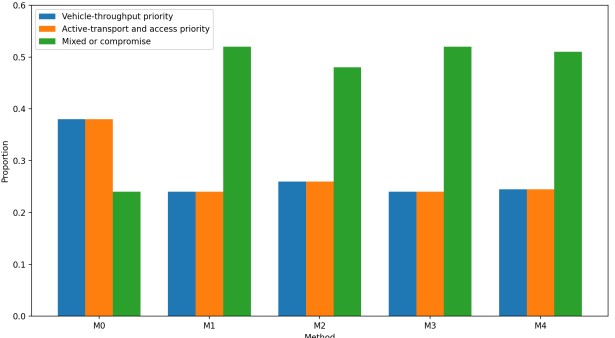

*Figure 2.* Outcome proportions by method (pilot). In this benchmark, the single-author baseline yields fewer mixed or compromise labels; curated and veto-enabled releases increase mixed or compromise and reduce dispersion across groups. The appropriate compromise rate is task-dependent.

coding constant. We compare five methods: single-author prompt (M0); one prompt sampled from the open commons (M1); prompts sampled from the curated commons under inclusion constraints (M2); curated prompts with veto-and-quarantine that excludes vetoed prompts (M3); and a negotiation ensemble (M4) sampling $k=6$ prompts stratified by author group with a versioned aggregation instruction (surface conflicts, propose a compromise, choose a label). Prompt sampling is performed with a fixed seed so that the full pipeline is reproducible.

We evaluate on a contested-choice benchmark of $N=50$ short vignettes about streetscape and public-space trade-offs. For each vignette, the model selects one of three domain-grounded labels: *vehicle-throughput priority* (emphasise private-vehicle throughput or parking), *active-transport and access priority* (emphasise walking, cycling, transit, or accessibility), or *mixed or compromise* (explicit trade-off, staging, or deferral). These labels are descriptive for this benchmark and are not intended as ideological categories. We report class proportions (Figure 2) and a task-specific commitment score $D = 1 - p_{\text{mixed}}$ as descriptive statistics.

For perceived acceptability, we use a blinded rating protocol with $R=12$ raters (two per self-identified group across six groups: seniors, women, ethnic minority, disability, LGBTQ+, religious minority). Each rater scores each output on a 7-point "acceptable for my group" scale. Outputs are presented in random order with method identity hidden. We report the mean of group means, where each group mean averages over its two raters and $N$ vignettes. Uncertainty is reported as Wilson intervals for class proportions over $N$ and as standard deviations over group means for acceptability.

For operational safety, we track time-to-remediation for harmful outputs flagged in an issue tracker with a defined workflow (report, quarantine, fix, close), measured as hours from report to logged resolution. In this pilot we use a synthetic incident arrival log with 50 flagged items per gov-

ernance state to isolate procedural effects, motivated by security analyses (Liu et al., 2023; Yao et al., 2024) and the OWASP Top 10 for LLM applications (OWASP GenAI Security Project, 2023). This component should be interpreted as a workflow stress test, not a field estimate of institutional response times.

### 4.3. Results

**Compromise rate and commitment.** The single-author baseline (M0) yields 38 % vehicle-throughput priority, 38 % active-transport and access priority, and 24 % mixed or compromise ($D=0.76$). Across open and curated commons (M1–M3), the mixed or compromise rate is 48 % to 52 % with the remainder split between the two objective-priority labels ($D \in [0.48, 0.52]$). Weighted ensembling (M4) yields $D=0.49$. These are descriptive proportions for this benchmark under fixed decoding and a fixed rubric, and they are reported to illustrate that prompt-governance choices can move output distributions.

**Subgroup satisfaction.** Across six self-identified groups, mean acceptability (7-point; mean of group means) is $4.35 \pm 0.86$ (M0), $4.92 \pm 0.44$ (M2), and $5.48 \pm 0.66$ (M3), where $\pm$ denotes standard deviation across group means. Dispersion across groups (Gini over group means) decreases from $0.096$ (M0) to $0.043$ (M2). These summaries are descriptive and are intended to be falsifiable under replicated rubrics, alternative task formulations, or alternative rater pools.

**Moderation efficiency.** In the synthetic incident log (50 flagged items per governance state), mean time-to-remediation under our workflow assumptions decreases from $30.5(89)$ h (open) to $11.8(32)$ h (curated) and $5.6(15)$ h (veto-enabled). These values illustrate that governance pro-

cedures can change operational response and connect to evidence that moderator support and response time shape experienced harms (Li et al., 2022; Cao et al., 2024; Tabassum et al., 2024).

## 4.4. Interpretation and limits

The pilot illustrates that prompt governance can be treated as an empirical object: prompt provenance can be represented in a repository, governance states can be implemented as workflows, and consequences tracked with task and operational metrics. It does not estimate general effect sizes. The observed mixed or compromise shift is specific to this deliberative benchmark; other tasks may require higher commitment or different objective trade-offs. We treat these metrics as task-specific descriptive statistics, not universal objectives.

The main limitations are that the pilot uses a single model accessed via API, a single benchmark of $N=50$ vignettes, and a small rater pool tied to one recruitment setting. The incident-response results use a synthetic arrival log. These choices make the pilot easy to reproduce and falsify, but they limit external validity.

The pilot also reflects a broader negotiation claim. If prompts are treated as transient inputs, disagreement about value framings occurs without a shared artefact to contest. By contrast, Prompt Commons makes disagreement legible as competing prompt proposals with recorded rationales, enabling negotiation over prompt text, aggregation rules, and deployment configurations.

# 5. Implications for Public-Sector Deployment

## 5.1. Governance realism: checklists and institutional constraints

Governance proposals must translate into implementable practices. Prompt Commons aims to make governance concrete by expressing rules as merge conditions and release criteria. In a public-sector deployment, a curated prompt collection can be treated as a release artefact that must pass a checklist before it can be used in an operational workflow. This mirrors established practices in software supply chains and content moderation.

To limit ambiguity in curation, the pilot uses a published checklist as a merge condition. It requires a locale tag; limits length; prohibits personally identifiable information; requires a stated value claim; requires accessibility tags when relevant; requires at least one counter-prompt for deliberative use; and requires an attached licence. In a public deployment, such constraints can be linked to procurement and oversight requirements by treating the checklist as part of the contract artefact. This approach is intended to re-duce silent drift in prompt templates while retaining local adaptability.

## 5.2. Procurement, vendor ecosystems, and licensing compatibility

Many public-sector deployments are mediated by vendors and integrators, and prompt templates move across contracts as implementation artefacts. When prompts are not treated as governed objects, accountability fragments: the provider controls weights and system policies, the vendor controls prompt templates and workflow glue, and the public institution inherits outputs without an audit trail. Prompt Commons supports procurement by making prompt configuration a citeable release object: documents can reference a prompt collection version, its governance state, and incident-response commitments.

Licensing affects whether prompt templates can be shared across agencies, whether vendors can incorporate community-authored prompts into commercial offerings, and whether derivative prompt sets remain open. For this reason, Prompt Commons treats licensing choice as a deployment parameter rather than a post hoc publication decision. This aligns with open-source governance experience in which licensing and process rules define the boundary between contribution and appropriation, and in which formal governance structures tend to emerge as stakeholder diversity and reuse increase (O'Mahony & Ferraro, 2007; The Linux Foundation, 2020).

## 5.3. Privacy, access control, and disclosure

Prompt repositories can contain sensitive material. A prompt may embed personally identifiable information, operational details about an agency, or instructions that could be misused if disclosed without context. This creates a governance tension: auditability pushes toward disclosure, while privacy, consent, and security push toward access control. Prompt Commons is compatible with staged disclosure in which prompt metadata (provenance, licence, change log, and withdrawal or veto records) are public, while prompt texts are restricted to authorised reviewers or released after redaction; this is a governance choice to be justified and audited, not a guarantee.

Security guidance for LLM applications emphasises separating untrusted inputs from control instructions and maintaining explicit incident-response procedures (Liu et al., 2023; Yao et al., 2024; OWASP GenAI Security Project, 2023). Prompt Commons extends this guidance by making prompt changes reviewable and by treating quarantine and rollback as first-class actions. Privacy and consent constraints can be operationalised: redaction as merge checks; restricted prompts stored encrypted with public version IDs; withdrawal requests remove prompts from future releases with

tombstone records (forks may persist).

### 5.4. Interoperability with model-level governance

Prompt Commons does not remove the need for model documentation and alignment. Model Cards and Datasheets provide model- and dataset-level information that is relevant for interpreting deployment risks (Mitchell et al., 2019; Gebru et al., 2021). Alignment methods constrain model behaviour across uses, but they do not determine how an institution frames a specific task (Christiano et al., 2017; Ouyang et al., 2022; Bai et al., 2022). Prompt governance can therefore be treated as an additional layer that should interoperate with model-level controls.

One implication is that prompt governance should record the model version and any provider-imposed system instructions that shape outputs. Another is that providers can support local governance by exposing interfaces for prompt provenance (stable identifiers, change logs, and policy constraints) without requiring access to weights. This separation also clarifies responsibility. Model providers retain responsibility for general model behaviour and disclosed limitations; public institutions and their communities retain responsibility for local framing choices that are expressed through prompt configuration.

### 5.5. From local commons to field-level benchmarks

A motivation for treating prompts as versioned artefacts is comparability. If prompt collections are released with provenance metadata and governance logs, researchers can compare deployment configurations across jurisdictions, vendors, and institutional mandates without requiring access to internal operational systems. This enables evaluation work that treats governance rules as variables, including how curation constraints change the distribution of outputs, how veto procedures affect incident response, and how aggregation prompts influence compromise proposals.

## 6. Negotiation, Aggregation, and Value Pluralism

Prompt Commons is motivated by value pluralism: public-sector deployments often involve competing value claims that cannot be resolved by a single notion of correctness. Rather than treating these conflicts as noise, Prompt Commons treats them as governance inputs. This motivates the inclusion of multiple prompt framings and the use of explicit aggregation prompts.

The aggregation method illustrated in the pilot (M4) uses an ensemble of stakeholder prompts and a negotiation-oriented instruction to propose a compromise. This can be interpreted as a prompt-level analogue of social-choice aggregation: the aggregation rule shapes which preferences and constraints are preserved in the final output (Huang et al., 2025; Conitzer et al., 2024). Prompt Commons makes the aggregation rule explicit and versioned, allowing it to be contested and revised.

Prompt Commons operationalises negotiation through a small set of repository artefacts. Prompt entries encode proposed framings and their provenance. Collections correspond to deployment releases that can be cited in procurement documents, evaluations, and incident reports. Aggregation prompts make the handling of disagreement explicit by specifying how competing prompts are combined in deliberative tasks. Issues and pull requests record arguments and counterarguments; veto records log harm claims that trigger quarantine; and withdrawal records log consent changes and remove prompts from future releases while preserving audit trails.

## 7. Evaluation Agenda and Research Directions

### 7.1. Evaluability and measurement

Treating prompts as governed artefacts creates new measurement opportunities. Beyond task-level output accuracy or subjective ratings, governance introduces operational metrics such as auditability, coverage, and responsiveness. Auditability includes whether a given output can be traced back to a specific prompt version and whether changes in outputs can be linked to changes in prompt text or governance rules. Coverage includes whether prompt releases represent a diversity of locales and stakeholder groups. Responsiveness includes how quickly harmful outputs can be addressed through revision and release workflows.

A research agenda for prompt-layer governance can study how different repository rules and moderation structures affect these metrics. For example, one can compare open versus curated versus veto-enabled governance in terms of prompt diversity, rates of harmful outputs, and time to remediate incidents. One can also evaluate negotiation-oriented aggregation prompts as procedures for combining value framings and managing disagreement.

### 7.2. Evaluation agenda and falsifiable claims

Position papers are expected to support claims with reasoning and evidence, and to be defendable against credible alternatives. Prompt Commons makes falsifiable predictions: (P1) with model, rubric, and inputs fixed, curated or veto-enabled releases shift output distributions and across-group dispersion; (P2) versioned releases improve auditability, measured by output-to-prompt traceability and reproducibility under logged configurations; (P3) quarantine and veto reduce remediation latency and repeat incidents under a specified workflow. The position is undermined if gov-

erned releases yield negligible shifts, no auditability gains, or higher harms or slower remediation under comparable resourcing.

These predictions suggest an evaluation agenda for the ML community. Treat governance rules (open versus curated versus veto-enabled) and aggregation prompts as experimental conditions, holding model, decoding, task, and inputs constant, and report both task metrics and governance metrics (coverage, veto rate, audit-log completeness). Compare aggregation prompts using social-choice criteria such as agenda control and sensitivity to minority objections (Huang et al., 2025; Conitzer et al., 2024). Integrate security evaluation by treating prompt changes as part of the attack surface and measuring the effectiveness of quarantine and review processes under simulated injection and misuse scenarios (Liu et al., 2023; Yao et al., 2024; OWASP GenAI Security Project, 2023).

## 8. Risks and Mitigations

### 8.1. Risk: Prompt capture and unequal participation

A prompt commons can be captured by well-resourced actors, leading to prompt sets that reflect dominant perspectives while claiming community legitimacy. This risk is shared with many participatory governance models. Prompt Commons mitigates capture by treating provenance fields and inclusion constraints as enforceable rules, by enabling minority veto in the veto-enabled state, and by making deliberation records public within the governance scope.

### 8.2. Risk: Over-caution and goal mismatch

Veto procedures and safety constraints can lead to over-cautious prompts that avoid substantive recommendations, potentially reducing usefulness in operational contexts. This motivates the use of task-specific evaluation and the separation of descriptive statistics from normative objectives. For example, a higher mixed or compromise rate in a deliberative benchmark may be desirable for public engagement but undesirable for emergency response tasks.

### 8.3. Risk: Security trade-offs

Prompt transparency can increase attack surface if prompts reveal system instructions or operational constraints. Prompt Commons mitigates this by enabling staged access control and by treating security review, quarantine, and rollback as governance primitives. This aligns with operational guidance for LLM application security (OWASP GenAI Security Project, 2023).

## 9. Alternative Views

Positioning prompt governance as a primary accountability surface competes with established views on where governance should be located in ML systems. We outline three credible alternatives in their strongest form and specify how each can be empirically distinguished from the Prompt Commons position.

**Alternative view 1: Model-level alignment should be the primary locus of governance.** Strongest version: governance should focus on training data, feedback procedures, and provider system policies because these constrain behaviour across use cases; if RLHF, instruction tuning, and constitutional approaches succeed, local prompt variation should have limited marginal effect and may not warrant separate governance (Christiano et al., 2017; Ouyang et al., 2022; Bai et al., 2022).

Response: Prompt Commons is compatible with model-level alignment but targets deployment-time framing choices that alignment cannot exhaustively specify. A procedural test is invariance: under a fixed aligned model and rubric, if outputs and harm rates are statistically indistinguishable across prompt governance states, then prompt-layer governance adds little; consistent, traceable shifts would support the prompt-layer claim.

**Alternative view 2: Prompt sets should remain closed to reduce manipulation and security risk.** Strongest version: prompt repositories should be treated as operational secrets because transparency can enable gaming, targeted jailbreaks, and prompt-injection attacks, and can expose sensitive workflow details (Liu et al., 2023; Yao et al., 2024; OWASP GenAI Security Project, 2023).

Response: Prompt Commons does not require full public disclosure; it requires governed access control with auditable provenance, change logs, and incident-response procedures. The falsifiable question is whether staged disclosure (public metadata, restricted texts) yields better auditability and comparable or lower attack success than ad hoc secrecy; if transparency measurably increases successful attacks or harms under comparable controls, the position recommends restricting disclosure, not abandoning governance.

**Alternative view 3: Compromise rate is not a coherent objective for policy decisions.** Strongest version: compromise is not a coherent objective for many policy decisions; higher mixed or compromise rates can be less actionable, can encode status quo bias, and can diffuse responsibility or mask distributional harms.

Response: Prompt Commons treats mixed or compromise rates and commitment as task-dependent descriptive statis-

tics, not normative goals; the core claim is procedural transparency and contestability. A falsifiable expectation is that task-mode declarations and evaluation rubrics determine whether governance shifts compromise rates up or down; systematic drift toward mixed outcomes across tasks would indicate mis-specified governance, not success (Huang et al., 2025; Conitzer et al., 2024).

## 10. Call to Action

For the ML community, Prompt Commons reframes prompt templates as governance artefacts that can be studied with the same rigour as datasets and models. For model providers, it suggests interfaces that support prompt provenance and audit logs. For public institutions, it suggests procurement and oversight practices that treat prompt configurations as citeable, governed releases rather than ad hoc inputs.

Concrete steps include: (1) developing benchmark tasks and metrics for prompt governance, including auditability and remediation metrics; (2) building tooling for versioned prompt repositories with provenance metadata; (3) studying aggregation prompts as procedural operators under social-choice criteria; and (4) integrating security evaluation and incident-response workflows into prompt governance.

## 11. Conclusion

Prompting is a control surface for LLM deployments, and in public-sector contexts prompt templates can circulate as de facto policy instruments. Existing governance artefacts rarely make local prompt collections transparent or contestable. Prompt Commons proposes a governance-oriented repository design for prompts, with enforceable rules, licensing, and moderation logs.

The pilot testbed illustrates how governance states can be operationalised and measured, but it does not claim general empirical validation. Its purpose is to demonstrate evaluable and falsifiable questions about prompt-layer governance. If governed prompt collections do not improve auditability, contestability, or incident-response performance relative to current practices, or if they introduce new harms that cannot be mitigated through access control and procedural design, then the position advanced here would be undermined.

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
