# OpenReview forum: "Position: Prompts for Public-Sector LLMs Should Be Governed as Commons"
_ICML.cc/2026/Position_Paper_Track — ICML 2026 Position Paper Track regular_

### Official Review · Reviewer_3PVF · 2026-03-12

**Significance:** 3
**Argument Clarity:** 2
**Rating:** 4
**Confidence:** 3

**Questions:**

Please see weaknesses

**Alternative Views Section:**

Yes

**Compliance With Llm Reviewing Policy A Conservative:**

Affirmed.

**Discussion Potential:**

2

**Final Justification:**

The rebuttal overall address my concerns. I decide to maintain my score.

**Paper Summary:**

This position paper argues that prompts for LLM deployments in the public sector should be governed as formal, shared commons rather than transient, private inputs. Specifically, the authors propose Prompt Commons, a versioned, community-maintained prompt repository with standardized provenance metadata, enforceable governance workflows, licensing rules, and audits.

**Position:**

Yes

**Position In Title:**

Yes

**Related Work:**

2

**Strengths And Weaknesses:**

Strengths:
1. This paper views prompt engineering from a technical optimization task to a governance practice, addressing the opaque, unregulated circulation of default prompts for LLMs.
2. The design is actionable, with empirical evidences.

Weaknesses:
1. The evaluation is small-scaled. It is not clear if the results are generalizable.

**Support:**

3

---

> ### Author Rebuttal · Authors · 2026-03-27
>
> Thank you for the constructive review and for recognizing both the reframing of prompt engineering as governance practice and the paper’s effort to make that framing actionable. We agree that the main question is empirical scope.
>
> The pilot is intentionally small and controlled, and the paper does not present it as evidence of general effect sizes across domains. Its role is narrower: to show that prompt governance can be operationalized, measured, and falsified as an independent variable. That is why the design holds model, inputs, rubric, and decoding fixed while varying governance state. Under those controls, the paper observes nontrivial shifts in output distributions, subgroup acceptability summaries, and remediation workflow. For a position paper, that is the key empirical point: the prompt layer is not merely rhetorical, but a measurable governance surface.
>
> The paper also states what would count against its own thesis. If governed releases produced negligible shifts under fixed conditions, no auditability gains, or slower remediation under comparable resourcing, the position would be weakened. In that sense, the pilot is designed less as a claim of external validity than as a demonstration of evaluability and falsifiability.
>
> We agree that broader validation across tasks, models, and institutions is the natural next step, but we view that as part of the research agenda the paper opens rather than a precondition for recognizing the contribution now. The paper’s main contribution is the combination of a governance claim, a concrete repository design with enforceable rules, and an empirical scaffold that makes the claim testable.
>
> To make this even clearer in revision, we would add a sentence such as: “The pilot should be read as feasibility evidence that prompt-layer governance is observable and measurable, not as a claim that the reported magnitudes generalize across domains.” For fuller discussion of scope and normative interpretation, we also point to our response to Reviewer mrKB.
>
> Thank you again for the thoughtful review.

---

> > ### Author Rebuttal · Reviewer_3PVF · 2026-04-02
> >
> > My concerns have been addressed and I decide to maintain my current score.

---

### Official Review · Reviewer_FLnd · 2026-03-12

**Significance:** 3
**Argument Clarity:** 4
**Rating:** 5
**Confidence:** 3

**Questions:**

1. How would Prompt Commons handle prompts for contested domains where no appropriate compromise exists (e.g., immigration enforcement, policing)?
2. The aggregation ensemble (M4) uses an LLM as the aggregation mechanism. How sensitive is this to the choice of aggregation prompt? Could this itself become a site of capture or manipulation?

**Alternative Views Section:**

Yes

**Compliance With Llm Reviewing Policy A Conservative:**

Affirmed.

**Discussion Potential:**

4

**Final Justification:**

I decide to keep my score after the retubbals.

**Paper Summary:**

This position paper argues that prompts used in public-sector LLM deployments constitute a distinct governance surface that existing mechanisms (Model Cards, Datasheets, post-training alignment, platform policies) do not adequately address. The authors observe that prompt templates act as de facto policy instruments—encoding role instructions, decision framings, and value claims—that can materially shift outputs even when model weights and inputs are held fixed. The paper proposes "Prompt Commons": a versioned, community-maintained repository of prompt templates with provenance metadata, licensing, and moderation logs. Three governance states are defined (open, curated, veto-enabled), adapted from Ostrom's commons governance principles and open-source practices. A pilot testbed is reported using 443 human-authored prompts (3,317 after augmentation) collected with community partners in a large North American city, evaluated on a benchmark of 50 streetscape/public-space vignettes. The pilot illustrates that governance states can shift output distributions (e.g., compromise rates) and subgroup satisfaction, with time-to-remediation decreasing under more structured governance. The paper also discusses negotiation-oriented prompt aggregation, licensing considerations, security tradeoffs, and interoperability with model-level governance.

**Position:**

Yes

**Position In Title:**

Yes

**Related Work:**

4

**Strengths And Weaknesses:**

## Strengths

- The identification of prompts as a distinct governance surface—separate from model weights, training data, and platform policies—is a genuine contribution.
- The paper provides a concrete repository design with enforceable rules, specific metadata fields, and defined governance workflows. The connection to Ostrom's commons principles and open-source governance is well-illustrated.
- The inclusion of a real pilot involving diverse community groups demonstrates practical engagement.

## Weaknesses

- The governance burden may be prohibitive. The proposed system requires provenance metadata, licensed contributions, merge checklists, veto mechanisms, quarantine procedures, and audit logs for prompt templates.

**Support:**

4

---

> ### Author Rebuttal · Authors · 2026-03-27
>
> Thank you for the positive assessment and for identifying the paper’s intended contribution so precisely. We are especially encouraged that you read the submission as making a genuine governance contribution. We address your three questions below.
>
> On governance burden: we agree that an unscoped prompt-governance process could become prohibitively heavy. The paper’s design is meant to avoid that by making governance graduated and release-based. The three states are not all-or-nothing obligations; they are risk-tiered options. Low-risk or exploratory uses can remain open with lightweight moderation. Higher-stakes deployments can move to curated or veto-enabled releases, where the extra process is justified because prompts are functioning as de facto policy instruments. Sections 5.1 to 5.3 are meant to make this realism concrete: merge checklists, inclusion constraints, staged disclosure, procurement references, quarantine, rollback, and withdrawal records. In that sense, Prompt Commons does not add governance on top of an otherwise clean baseline; it makes existing prompt edits, which already happen across staff and vendors, visible and accountable. The pilot’s remediation analysis illustrates the practical upside of that structure: under the specified workflow, stronger governance reduced time-to-remediation.
>
> On contested domains where no appropriate compromise exists, we fully agree that “finding the middle” can be normatively wrong. Prompt Commons is not committed to compromise as the desired endpoint. Its commitment is to explicit governance of disagreement. In some domains, that may mean maintaining separate releases for incompatible mandates, enabling protected veto, imposing rights-based exclusions, or deciding that an LLM should not be deployed for the task at all. Policing and immigration enforcement are exactly the kinds of cases where disagreement may be irreducible and where procedural legitimacy may require non-aggregation rather than synthesis. We see this as an important strength of the framework: it does not force a false consensus, but records where consensus fails and what governance action follows.
>
> On M4 and sensitivity to the aggregation prompt: yes, absolutely. The aggregation prompt can itself become a site of capture, manipulation, or agenda control. That is precisely why the paper includes it as a first-class governed artefact rather than treating it as a hidden implementation detail. In Prompt Commons, the aggregation rule is versioned, reviewable, and contestable alongside ordinary prompts. This lets institutions compare alternative aggregation prompts, attach provenance and rationale to them, and evaluate them under social-choice style criteria such as sensitivity to minority objections or agenda dependence. In other words, the possibility of capture is not an objection to making aggregation explicit; it is a reason to govern it.
>
> To make these points even more explicit in revision, we would add language such as: “Prompt Commons does not assume compromise is always appropriate; in domains with rights-based boundaries or irreducible conflict, governance may instead require protected veto, separate releases, or non-deployment.” We would also add: “Because aggregation prompts can shape outcomes, they are themselves governed artefacts subject to versioning, audit, and contestation.”
>
> Thank you again for the strong and careful review.

---

> > ### Author Rebuttal · Reviewer_FLnd · 2026-04-03
> >
> > Thank the authors for the rebuttal which largely resolves my concerns. I will maintain my positive score.

---

### Official Review · Reviewer_mrKB · 2026-03-13

**Significance:** 3
**Argument Clarity:** 3
**Rating:** 3
**Confidence:** 3

**Questions:**

Please refer to the weakness part.

**Alternative Views Section:**

Yes

**Compliance With Llm Reviewing Policy A Conservative:**

Affirmed.

**Discussion Potential:**

2

**Final Justification:**

My concerns are not fully addressed. The paper needs to be revised further to address my concerns. However, I'm not going to be offended if the paper is accepted. It is a 'weak reject' after all.

**Paper Summary:**

The paper argues that prompts used in public-sector LLM deployments should be treated not as incidental inputs, but as governable public artifacts. Its central position is that prompt collections shape downstream decisions, values, and trade-offs in ways that existing model-level governance tools—such as model cards, datasheets, and provider-side alignment—do not adequately expose or regulate at deployment time. The authors therefore advocate making prompts visible, versioned, contestable, and auditable as part of public accountability infrastructure.

The paper’s main contribution is the proposal of Prompt Commons, a governance framework for managing prompts as formal deployment artifacts. Under this framework, prompts are stored in a versioned repository together with provenance, licensing, locale, value claims, accessibility tags, revision history, audit logs, and withdrawal records. The framework also defines three governance regimes—Open, Curated, and Veto-enabled—that specify progressively stronger review, metadata, and quarantine procedures for deciding which prompts can enter official releases. In this sense, the paper contributes a concrete institutional design rather than merely a high-level call for “better prompting.”

To make the position empirically discussable, the paper also presents a pilot testbed. The authors compile a repository of community-authored prompts from a North American city context, expand these prompts into a larger pool, and compare several prompt-governance settings: a single-author prompt baseline, open commons sampling, curated commons sampling, veto-enabled curation, and a negotiation-style ensemble that aggregates prompts from multiple groups. Using a fixed chat LLM and a benchmark of 50 urban-policy vignettes, they examine how different governance setups shift output distributions, subgroup acceptability scores, and remediation time in a synthetic incident-response workflow. The paper does not claim to prove universal superiority, but instead uses this pilot to support the broader position that prompt governance is an operational, measurable, and contestable layer of AI governance in its own right.

Overall, the paper is best understood as a position paper with an accompanying governance design and illustrative empirical scaffold. Its advocated position is that public-sector AI governance should expand beyond models and datasets to include prompt repositories, release procedures, and prompt-level accountability mechanisms, especially in settings where multiple communities and value claims are implicated in deployment.

**Position:**

Yes

**Position In Title:**

Yes

**Related Work:**

3

**Strengths And Weaknesses:**

Strength
1. A major strength is the choice of problem and framing. Much of the current LLM governance literature focuses on model training, datasets, alignment, and platform-level safety policies. This paper usefully shifts attention to prompts as a distinct deployment-time governance surface. That framing is insightful because, in practice, prompt templates often circulate informally across teams, vendors, and contractors, while their embedded assumptions are easily mistaken for properties of the model itself rather than contingent choices made by local actors. The paper makes a persuasive case that prompt configuration can shape the considerations that appear relevant, how uncertainty is communicated, and which trade-offs are presented as admissible in public-sector settings.

2. A second major strength is that the paper does not remain at the level of abstract normativity. Instead, it specifies Prompt Commons as a fairly concrete governance design with recognizable operational components: versioned repositories, provenance metadata, licensing, accountable moderation, merge and review procedures, governance states, incident-response commitments, and auditable release objects. The discussion of procurement, vendor mediation, privacy, staged disclosure, access control, quarantine, rollback, and withdrawal records makes the position stronger. This is important because many AI governance papers identify valid concerns but stop short of offering implementable mechanisms. Here, the authors articulate a workflow that others could critique, modify, or test.

3. The authors do not oversell the pilot as proving universal effectiveness. They explicitly present it as an illustrative testbed and state testable predictions rather than claiming that Prompt Commons has already been validated across contexts. In the results section, they are similarly careful to frame shifts in label proportions as descriptive statistics under a fixed benchmark, used to show that prompt-governance choices can move output distributions.

Weakness
1. One major weakness is that, while the paper’s core position is clear, the thesis is not yet as sharp or firm as it could be for a strong Position Track submission. The authors argue that prompts in public-sector LLM deployments should be treated as governable public artifacts rather than incidental inputs, and this is a valid and interesting claim. However, the paper is less convincing on the stronger question of why the prompt layer should be elevated as a particularly urgent or primary governance target relative to other established intervention, such as models, datasets, alignment procedures, documentation, or procurement oversight. As written, the argument sometimes reads more like an appeal to add one more layer of governance than a decisive case that current governance frameworks systematically fail without prompt-level institutionalization.

2. Another main weakness is the limited empirical foundation. The pilot is useful as a proof of evaluability, but it remains narrow: a fixed-seed setup, a single benchmark of 50 short vignettes, 12 raters across six groups, and a synthetic incident log for remediation analysis. The paper itself notes that the operational-safety component should be interpreted as a workflow stress test rather than a field estimate of institutional response times. As a result, the empirical results are better read as feasibility evidence than as strong validation of the proposed governance framework.

3. A related weakness is that some of the reported effects are difficult to interpret normatively. The paper is careful to say that increased mixed/compromise outputs are task-dependent and descriptive rather than inherently desirable, which is the right caveat. Still, this caution also reveals a limitation: the pilot demonstrates that governance choices move outputs, but it does not fully establish when those movements are substantively better, for whom, and under what decision regimes. In other words, the paper succeeds at showing sensitivity to governance state, but only partially succeeds at grounding a normative criterion for why a given shift should count as governance improvement beyond subgroup acceptability and dispersion measures.

4. Another weakness is that the paper’s strongest contribution is primarily institutional framing and design, not technical novelty in the usual ML sense. That is not necessarily disqualifying for a Position Track paper, but it does make the submission vulnerable to reviewers who expect a more technically developed systems or methodological contribution. The pilot measures consequences of governance regimes, yet it does not provide a deep empirical comparison across models, domains, or deployment settings, nor does it offer a formal framework for optimizing or verifying these governance choices.

**Support:**

2

---

> ### Author Rebuttal · Authors · 2026-03-27
>
> Thank you for the careful and constructive review. We are encouraged that you see the problem choice, the deployment-time framing, and the repository design as the paper’s core strengths.
>
> Our thesis is not that prompt governance replaces model, data, alignment, or procurement governance. It is that these mechanisms are incomplete without prompt-layer governance, because the prompt layer is where local institutions operationalize normative discretion under a fixed model. In many public-sector procurements, agencies do not control model weights at all, so prompt configuration is often the main local lever they do control. In public-sector deployments, prompts are frequently edited by staff, contractors, or vendors; they encode role, audience, decision framing, and admissible trade-offs; and they often circulate outside formal review. That is why the prompt layer is especially urgent: it is the smallest, most mutable, and least audited site where local policy choices become system behavior. The paper’s identification strategy is designed to show precisely this independence: with model, inputs, rubric, and decoding fixed, changing the prompt governance state changes output distributions, subgroup acceptability, and remediation workflow. That is the decisive reason prompt governance is not merely “one more layer.”
>
> On empirical scope, we agree with your description of the pilot as feasibility evidence rather than a universal field estimate. That is exactly the intended role. This is a position paper with a concrete institutional design and an evaluable empirical scaffold. The pilot is not meant to estimate cross-domain effect sizes; it is meant to show that the position yields falsifiable predictions and measurable consequences. The paper states three such predictions explicitly: governance should shift output distributions under fixed models and inputs; versioned releases should improve traceability and reproducibility; and quarantine or veto procedures should change remediation latency under a specified workflow. A narrow, controlled pilot is appropriate for that claim because it tests whether the prompt layer is observable and governable as an independent variable at all. If governed and unguided prompt releases produced no measurable differences under fixed conditions, the thesis would be substantially weakened.
>
> On normative interpretation, we agree that “more mixed or compromise outputs” is not a universal good. The paper is intentionally careful on this point. The compromise rate and commitment score are descriptive statistics for this deliberative benchmark, not normative objectives for public policy generally. Governance improvement in the paper is defined first procedurally: contestability, auditability, traceability, coverage, and response capacity. Subgroup acceptability and dispersion are task-specific indicators of whether a release is broadly tolerable across affected groups. In domains that require decisive action, or in domains where rights-based boundaries preclude compromise, a drift toward mixed outputs would count against a prompt release, not in its favor. We see this as a strength of the paper’s normative stance: it does not smuggle in a single welfare function for plural public contexts, but instead makes task mode, evaluation rubric, and aggregation rule explicit and contestable.
>
> On novelty, we agree that the contribution is not “technical novelty in the usual ML sense,” and we view that as appropriate for this track. The novelty is institutional and evaluative: the paper identifies prompt collections as a distinct governance object, specifies repository-enforceable primitives (versioned releases, provenance, licensing, veto, quarantine, withdrawal, audit logs), and turns governance states and aggregation prompts into variables that can be compared empirically. That combination moves the literature from a general call for better prompting or transparency to a concrete, testable governance framework.
>
> To make this framing even more explicit in revision, we would add language such as: “Prompt Commons does not claim that prompts are the only or even always the dominant governance surface; it claims that without governance of deployment-time prompt configuration, model-level and procurement-level controls cannot explain, reproduce, or contest local framing choices made under a fixed model.” We would also add: “In this paper, governance improvement is assessed procedurally, through auditability, contestability, and remediation capacity; shifts toward compromise are benchmark-dependent descriptive effects, not universal normative targets.”
>
> We hope this clarification addresses the reasons for your score. We appreciate the seriousness of your review and think your concerns sharpen, rather than undercut, the paper’s contribution.

---

> > ### Author Rebuttal · Reviewer_mrKB · 2026-04-04
> >
> > Thank you for rebuttal. My concerns and questions are partially resolved. However, I wish to see a revised version of the paper that directly addresses these concerns, e.g. on the empirical foundations and the normative interpretation. Thus, without the ability to review them, I see that the authors should take this opportunity to revise the paper further for a future submission. My score will remain the same. Thank you for engaging in the rebuttal period.

---

### Official Review · Reviewer_qxAX · 2026-03-23

**Significance:** 2
**Argument Clarity:** 2
**Rating:** 4
**Confidence:** 3

**Questions:**

See Weaknesses

**Alternative Views Section:**

Yes

**Compliance With Llm Reviewing Policy A Conservative:**

Affirmed.

**Discussion Potential:**

2

**Paper Summary:**

This paper proposes a governance framework called Prompt Commons, aimed at addressing the lack of transparency and reproducibility of prompts in public-sector deployments of large language models (LLMs). The authors argue that prompts should not be treated as transient inputs, but rather as governed artifacts that require systematic management. The framework establishes a community-maintained repository with version control, provenance metadata, licensing mechanisms, and audit logs.

**Position:**

Yes

**Position In Title:**

Yes

**Related Work:**

2

**Strengths And Weaknesses:**

It effectively identifies the "prompt layer" as a critical gap in existing AI accountability structures.

The paper reframes prompting as a negotiation interface for competing values. This is a strong conceptual contribution.

================================================

The evaluation relies on a relatively small set of 50 vignettes and 12 raters.  While the authors acknowledge this as an illustrative testbed, the generalizability to high-stakes or more complex policy domains remains unproven.

The results show a significant shift toward "mixed or compromise" outputs. While beneficial for deliberation, the paper should further clarify when such a shift might be undesirable

Issues such as governance overhead, stakeholder conflict, institutional constraints are not fully addressed.

**Support:**

2

---

> ### Author Rebuttal · Authors · 2026-03-27
>
> Thank you for the careful review. We are encouraged that you view the paper’s central contribution, identifying the prompt layer as a missing accountability surface and reframing prompting as a negotiation interface for competing values, as a strong conceptual contribution. We address your three concerns directly.
>
> First, on the pilot’s scale and generalizability: we agree that the 50-vignette, 12-rater pilot is not a field estimate for all high-stakes public deployments. The paper is intentionally explicit on this point. Its empirical role is narrower and, we believe, appropriate for a position paper: to show that prompt governance is evaluable as an independent variable, not to claim universal effect sizes. The key identification choice is that model, inputs, rubric, and decoding are held fixed while governance state varies. Under those controls, governed releases shift output distributions, group acceptability summaries, and remediation workflow. That is the core empirical support the position needs. The paper also states what evidence would undermine the thesis: negligible distributional shifts, no auditability gains, or slower remediation under comparable conditions.
>
> Second, on the increase in “mixed or compromise” outputs: we agree this can be undesirable in some tasks. The paper does not treat compromise rate as an objective to maximize. It treats compromise rate and commitment score as descriptive, benchmark-specific statistics. In a deliberative urban-planning benchmark, more explicit trade-off surfacing may be useful. In emergency response, benefits adjudication, or rights-sensitive domains, a drift toward compromise could be a failure mode. The governance contribution is therefore not “Prompt Commons produces more compromise,” but rather “Prompt Commons makes the framing rule, aggregation rule, and release process explicit, versioned, and contestable.” That explicitness is precisely what allows institutions to declare that certain tasks require decisive outputs, or that certain domains should not use aggregation at all.
>
> This connects to your third point about governance burden, stakeholder conflict, and institutional constraints. Open, curated, and veto-enabled states let institutions match procedural strength to deployment risk. Sections 5.1 to 5.3 already ground this in concrete mechanisms: merge checklists, inclusion constraints, procurement ties, staged disclosure, access control, quarantine, rollback, and withdrawal records. In other words, the proposal is not calling for new heavy bureaucracy; it translates governance into repository-enforceable release conditions. Importantly, the pilot’s remediation analysis also shows why some overhead is productive rather than merely costly: structured governance reduced time-to-remediation relative to open handling under the specified workflow assumptions. The practical comparison is not “Prompt Commons versus zero work,” but “visible, accountable process versus today’s hidden prompt drift across staff, vendors, and contractors.”
>
> For contested domains where no appropriate compromise exists, Prompt Commons does not require consensus-seeking. It can support separate prompt releases, rights-based exclusions, veto and quarantine, or a decision not to deploy an LLM for that task. We see that as a strength of the framework: it makes irreducible disagreement legible instead of laundering it into a single default prompt.
>
> To make these points even clearer in revision, we would add language such as: “The pilot is designed to establish evaluability and falsifiability of prompt-layer governance, not to estimate general effect sizes across domains.” We would also add: “Compromise-oriented aggregation is task-conditional; in domains with rights-based constraints or no legitimate middle ground, governance may require non-aggregation, protected veto, or non-deployment rather than compromise.”
>
> We hope this addresses the reasons for your borderline score. For the broader scope and normative-criterion discussion, we also point to our response to Reviewer mrKB.

---

> > ### Author Rebuttal · Reviewer_qxAX · 2026-04-05
> >
> > Thank the authors for the response!

---

### Decision · Program_Chairs · 2026-04-30

**Decision:**

Accept (regular)

**Comment:**

There is consensus among reviewers that the selection of problem and framing of position reflect sufficient novelty that this position will be of interest to the ICML community. While the technical community itself is not best-positioned to instrument the position itself, the overlapping connection points between the technical community and public-sector may be sufficientl high due to the size of the tecnical community that seeing this idea will eventually percolate to public-sector institutions. Given concerns that reviewers highlight regarding a need to sharpen the technical formulation, it may incidentally be positive that such a position is first presented to the technical community to see refinement before evaluation by the institutions that would ultimately contribute.

While this position diverges in topic from many that were submitted, it is worth consideration for inclusion primarily based on the novelty of its argument in relation to the current foci of model governance.